# Coarse Woody Debris Decomposition Assessment Tool: Model validation and application

Zhaohua Dai[1,2]*, Carl C. Trettin[1], Andrew J. Burton[2], Martin F. Jurgensen[2], Deborah S. Page-Dumroese[3], Brian T. Forschler[4], Jonathan S. Schilling[5], Daniel L. Lindner[6]

1 Center for Forested Wetlands Research, USDA Forest Service, Cordesville, SC, United States of America, 2 College of Forest Resources and Environmental Science, Michigan Technological University, Houghton, MI, United States of America, 3 Rocky Mountain Research Station, USDA Forest Service, Moscow, ID, United States of America, 4 Department of Entomology, University of Georgia, Athens, GA, United States of America, 5 Plant & Microbial Biology, Itasca Biological Station & Laboratories, University of Minnesota, Saint Paul, MN, United States of America, 6 Northern Research Station, USDA Forest Service, Madison, WI, United States of America

* zhaohuad@mtu.edu

**Data Availability Statement:** All relevant data are within the manuscript and its Supporting Information files.

## Abstract

Coarse woody debris (CWD) is a significant component of the forest biomass pool; hence a model is warranted to predict CWD decomposition and its role in forest carbon (C) and nutrient cycling under varying management and climatic conditions. A process-based model, CWDDAT (Coarse Woody Debris Decomposition Assessment Tool) was calibrated and validated using data from the FACE (Free Air Carbon Dioxide Enrichment) Wood Decomposition Experiment utilizing pine (*Pinus taeda*), aspen (*Populous tremuloides*) and birch (*Betula papyrifera*) on nine Experimental Forests (EF) covering a range of climate, hydrology, and soil conditions across the continental USA. The model predictions were evaluated against measured FACE log mass loss over 6 years. Four widely applied metrics of model performance demonstrated that the CWDDAT model can accurately predict CWD decomposition. The $R^2$ (squared Pearson's correlation coefficient) between the simulation and measurement was 0.80 for the model calibration and 0.82 for the model validation ($P<0.01$). The predicted mean mass loss from all logs was 5.4% lower than the measured mass loss and 1.4% lower than the calculated loss. The model was also used to assess the decomposition of mixed pine-hardwood CWD produced by Hurricane Hugo in 1989 on the Santee Experimental Forest in South Carolina, USA. The simulation reflected rapid CWD decomposition of the forest in this subtropical setting. The predicted dissolved organic carbon (DOC) derived from the CWD decomposition and incorporated into the mineral soil averaged 1.01 g C m$^{-2}$ y$^{-1}$ over the 30 years. The main agents for CWD mass loss were fungi (72.0%) and termites (24.5%), the remainder was attributed to a mix of other wood decomposers. These findings demonstrate the applicability of CWDDAT for large-scale assessments of CWD dynamics, and fine-scale considerations regarding the fate of CWD carbon.

**Funding:** Funder Name: U.S. Department of Energy (DOE, USA) Grant Number: DE-SC0016235 Grant Recipient: Dr Carl C Trettin Funder Name: National Science Foundation (NSF, USA) Grant Number: DEB 1754603 Grant Recipient: Dr Andrew J Burton Funder Name: Natural Science Foundation (NSF, USA) Grant Number: DEB 1754616 Grant Recipient: Dr Jonathan S Schilling.

**Competing interests:** The authors have declared that no competing interests exist.

## Introduction

Coarse wood debris (CWD) is a significant forest carbon (C) pool [1, 2], with the global input rate of CWD ranging from about 0.12 to 30.0 Mg ha$^{-1}$ yr$^{-1}$ [2]. Decomposition of CWD plays an important role in nutrient and C cycling in forests [3–8]. Correspondingly, the stock of CWD within a forest is dependent on the turnover rate [9–11], which is dependent on many factors, including decomposer community, temperature, moisture, and wood properties [2, 12–17].

A variety of empirical models, including single-exponential [18–20], multiple-exponential [21], linear [3], and lag-time [2] have been developed in attempts to characterize CWD decomposition using interactions among wood properties and abiotic factors. However, broad inferences from predictions based on empirical models are constrained by the measurements taken at study sites or regions that were used to develop the predictive equations. In contrast, mechanistic models can integrate fundamental processes using forcing functions based on long-term measurements and experiments, hence providing more robust applicability.

Russell et al. [22] utilized CWD surveys conducted in conjunction with forest inventory plots to effectively estimate CWD decomposition by assessing transitions among decay classes in the eastern United States. Further, several models have been developed to assess CWD decomposition without relying entirely on empirical relationships. Yin [23] developed a numerical model, based on the methodology used to analyze the long-term dynamics of C and nitrogen (N) in forest soils, as suggested by Agren and Bosatta [24]. That model incorporated substrate quality and microbial activity that was sensitive to abiotic conditions. Zell et al. [25] incorporated meta-analysis and a mixed effects model to predict CWD decomposition for 42 species that was sensitive to abiotic conditions. Although these models are not fully mechanistic, they are better than simple empirical models. The process-based soil C model, Yasso, added functionality to assess woody litter decomposition in soils and was tested using litterbag data in Canada [26, 27], and subsequently was used as the foundation for assessing CWD decomposition in boreal forests [28].

When considering the role of CWD decomposition with respect to the various forest C pools, environmental fluxes across habitats, and a changing climate, it is necessary to use a mechanistic model that reflects inherent biogeochemical processes. The CWDDAT (Coarse Woody Debris Decomposition Assessment Tool) has been developed to simulate CWD decomposition in forests by incorporating organic matter decomposition processes and ecological drivers regulating CWD mass loss as well as the fate of the wood C for climatic conditions that range from the tropics to the boreal zone [29].

The main functionalities of this model have been tested using different ambient conditions from 89 sites (14° N to 65° N latitude and 58 to 139° W longitude) distributed from the tropics to boreal zones, with a large range in mean annual temperature (-11.8° to 26.5° C), annual precipitation (181 to 6,143 mm), annual snowfall (0 to 612 kg m$^{-2}$), and elevation (3 m to 2,824 m above mean sea level). The model has also been tested with respect to wood properties, including species group (hardwood and softwood), wood density (0.1 to 1.0 g cm$^{-3}$) and size (4.0 to 45 cm in diameter). The results from the model assessment showed that CWD decomposition was highly sensitive to climatic factors, elevation, CWD properties and position (standing vs downed) [29]. Accordingly, the assessment affirms that CWDDAT should be a potentially effective tool for estimating CWD dynamics in forests.

Here we report on: (1) model calibration using observations from four sites to affirm whether or not the model performs within the design objectives [29]; (2) model validation using observations from five sites to assess whether or not the model performs as well with different observational inputs as it did under testing and calibration conditions, and (3) model

application by assessing decomposition of CWD left by Hurricane Hugo on the Santee Experimental Forest (Santee EF) in South Carolina in 1989. To achieve the first two objectives the observations from nine sites on the FACE (Free Air Carbon Dioxide Enrichment) Wood Decomposition Experiment (FWDE) are used to provide decomposition response data (6 years) for three tree species. The assessment of CWD following Hurricane Hugo is predicated on pre- and post-storm forest assessments which encompass eight decades [30–33].

## Materials and methods

### Study sites

The FWDE utilized nine Experimental Forests (EF) representing different bio-climatic zones to study CWD decomposition across the continental USA (Fig 1, Table 1) [34]. The study sites range in mean annual precipitation from 576 mm at the Fraser EF in Colorado to 2,259 mm at the Coweeta EF in North Carolina based on local climatic observations from 2000 to 2017 (S1 Table). The lowest annual mean temperature of 2.2˚C is at the Tenderfoot EF in Montana and the highest temperature of 18.6˚C is at the Santee EF site in South Carolina (Table 1). The elevation ranges from 8 m above mean sea level at Santee EF to 2,710 m at Fraser EF. There are substantial differences in vegetation coverage among sites, ranging from sparse shrub cover at the San Dimas EF in California to intact forests at Coweeta in North Caronia and Caspar Creek EF in California, leaf area index (LAI) ranges from 0.1 to 5.5 $m^2$ $m^{-2}$ (Table 1), with an average of 2.6 $m^2$ $m^{-2}$.

The Santee EF is located in South Carolina of USA (33.15˚ N, 79.8˚ W); it includes a paired watershed system [33], comprising a reference watershed (WS80) and a treatment watershed (WS77). Hurricane Hugo, a Category IV hurricane (1989), severely impacted the Santee EF. Over 80% of trees within WS80 were broken or uprooted, about 130 Mg $ha^{-1}$ of CWD were left in the watershed, based on the post hurricane inventory [30, 35]. All CWD produced by the hurricane on WS80 were left *in situ* without salvage logging. The Santee EF has biotic and abiotic observations over the last 80 years, including climate, biomass, and hydrology, which are useful for evaluating CWDDAT model applications; the forest is also an area with the highest prevalence of subterranean termites [36].

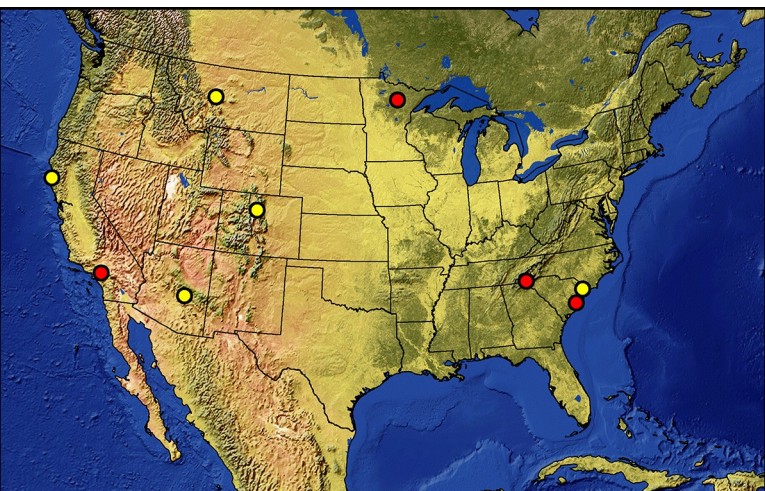

**Fig 1. Nine FACE wood decomposition experiment study sites in the continental USA used for model calibration and validation; red spots are the sites for model calibration, and yellow for the validation; this figure was produced with the data from Natural Earth.** Free vector and raster map data @ naturalearthdata.com.

**Table 1. Locations of FACE wood study sites used for model calibration and validation\*.**

| Site | State | Latitude (°N) | Longitude (°W) | Altitude (m) | MAT (°C) | MAP (mm) | Soil Series | Forest Type | Use | LAI |
|------|-------|---------------|----------------|--------------|----------|----------|-------------|-------------|-----|-----|
| Caspar Creek | CA | 39.3726 | 123.7063 | 240 | 10.8 | 1175.4 | Vanda-mme | Douglas fir | VL | 3.5 |
| Coweeta | NC | 35.0467 | 83.4574 | 910 | 12.2 | 2,259.2 | Evard-Cowee | Oak-hickory | CL | 5.5 |
| Duke | NC | 33.9760 | 79.0924 | 170 | 15.8 | 1,434.8 | Appling | Loblolly-shortleaf pine | VL | 2.5 |
| Fraser | CO | 39.9296 | 105.8698 | 2710 | 2.9 | 576.4 | Herd-Frisco | Fir-spruce | VL | 2.5 |
| Marcell | MN | 47.5057 | 93.4861 | 430 | 3.7 | 670.6 | Cutaway | White-red-jack pine | CL | 2.2 |
| San Dimas | CA | 34.2064 | 117.7615 | 670 | 17.7 | 523.1 | Trigo | Sparse Shrub | CL | 0.1 |
| Santee | SC | 33.1482 | 79.7910 | 8 | 18.6 | 1,673.9 | Wahee | Loblolly-shortleaf pine | CL | 2.5 |
| Sierra Ancha | AZ | 33.8039 | 110.9159 | 2220 | 11.2 | 717.8 | Sobega | Ponderosa pine | VL | 2.2 |
| Tender-foot Creek | MT | 46.9236 | 110.8697 | 2130 | 2.2 | 857.8 | Jeff-lake-Yellow-mule-Lonniebee | Lodgepole pine | VL | 2.2 |

\*: Use: CL, calibration; VL, validation; MAT, mean annual temperature; MAP, mean annual precipitation, based on the local climatic data observed between 2011 and 2017 (S1 Table); LAI, leaf area index, m² m⁻².

This work utilizes observations provided by the FWDE [34], which is located on the USDA Forest Service, Experimental Forest and Range Network [37]. Details regarding the FWDE and individual sites are provided by Trettin et al. [34]. Several of the FWDE primary investors (PIs) are U.S. Forest Service employees (Drs. Trettin, Page-Dumroese and Lindner), and they thus are able to request use of the Experimental Forest lands. We do recognize the Experimental Forest managers we have worked with in the acknowledgements. Data supporting the analyses of the post Hurricane Hugo response of CWD on the Santee EF were obtained from the cited literature [30–33, 35].

## Observation setup

Three species were used for the FWDE: loblolly pine (*Pinus taeda* L.) from the Duke Forest FACE site in North Carolina, USA, and aspen (*Populus tremuloides* Michx.) and birch (*Betula papyrifera* Marshall) from the Rhinelander FACE site in Wisconsin, USA. Twenty-four aspen and birch logs (1 m in length) and 24 pine logs (2 m in length) were placed horizontally on the surface of the forest floor to represent downed CWD (Fig 2A). Also, 24 loblolly pine logs (2 m in length) were placed vertically to emulate standing dead snags by placing the logs upright on

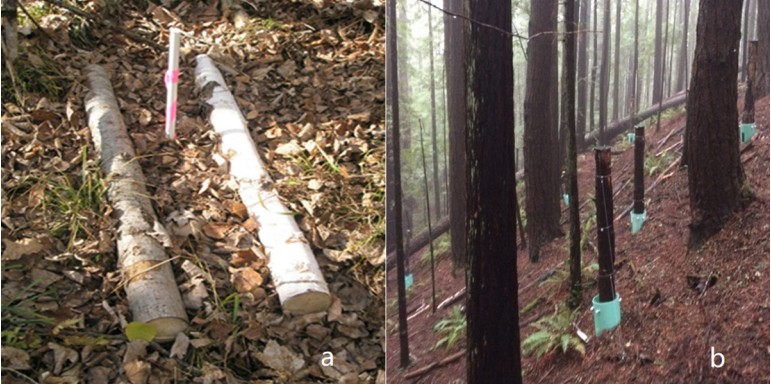

**Fig 2. Horizontal (downed) FACE logs.** (a) and vertical (standing) FACE woods (b).

a PVC pipe in a manner to preclude ground contact (Fig 2B). The fresh logs following harvesting were deployed in 2011 [34], for this study we utilize response data following 6 years decomposition. Those FACE logs used for this experiment had intact bark.

## Sample collection and analysis

A 3 to 4 cm thick disk was cut from the end of each log before they were installed in EFs to measure their initial wood density at time zero ($T_0$). The weight and length of each log and diameter at each end and in the middle of each log were measured, and the dry weight, volume, and surface area for each log were calculated. Initial wood density was measured by water immersion followed by drying (105°C) and weighing [34]. In the sixth year ($T_6$) following installation, each log was weighed in the field, unless it was too rotten to be handled; two disks (3–4 cm) were also cut from each end of the log, weighed *in situ*, and immediately placed in a plastic bag and sealed for transport to the laboratory. The $T_6$ disk samples were used for density determination by immersion in water to determine volume, followed by drying at 105°C. The moisture content of the disk samples was used to adjust the log field weight to a dry weight basis. The $T_6$ wood density and calculated log volume were also used to calculate a log weight. The moisture content and wood density averaged from the two disks collected from a FACE log *in situ* were used to calculate the wood mass loss for the log. The measured log mass loss (MLML) was the difference in log dry weight between the time $T_0$ and $T_6$. The calculated log mass loss (CLML) was based on the difference in log mass based on the log volume and wood density of the disks between the time $T_0$ and $T_6$.

## Model calibration and validation

Calibration and validation are important steps in applying a numerical model [38], since (1) model calibration is used to determine whether or not the model is in line with the design and application objectives, and (2) model validation is used to determine whether or not the model is performing as expected based on measurements. Accordingly, a well-performing model will produce similar results in the calibration and validation steps, thereby affirming applicability of the model for the target study. However, the datasets for the model validation should be completely different from those used for calibration. Accordingly, the CWDDAT was calibrated using observations from four sites with a range in climate, elevation and latitude; and the model was validated using observations from the other five sites (Table 1) with varying ecological conditions among the sites. The simulation results were compared to the year 6th FWDE observations of mass loss from the FACE logs.

## Model parameterization

The CWDDAT was parameterized using climate, soil, and vegetation data from the EF sites (Table 1). Climate data were obtained from weather stations near the EFs (S1 Table) and the Daymet database [39] for fill-in missing climatic observations. The $T_0$ data for each EF site was were summarized for the model parameterization (Table 2). The logs were divided into different size classes based on the measurements at the time $T_0$; The FACE logs were divided into two species groups: hardwoods–aspen and birch (coded as species 1, Table 2) and softwood–loblolly pine (coded as species 2, Table 2).

Assumptions for the simulation were: (1) annual litter fall for each site was initialized based on current forest species and biomass, this is used because litter can cover downed logs thereby affecting moisture content and temperature, (2) LAI (leaf area index) was initialized for each site based on the current forest type because leaf area can affect soil water content and temperature by changes in throughfall and light radiation to the forest floor, and (3) aboveground

**Table 2. Parameters of FACE logs used to calibrate and validate the model**[*]**.**

| Location | Position | Species | Size | Diameter (cm) | Log Mass (kg) | Wood Density (g cm$^{-3}$) |
|---|---|---|---|---|---|---|
| Caspar Creek | Down | 1 | 1 | 5.96 | 55.35 | 0.47 |
| | | 1 | 2 | 8.76 | 35.74 | 0.47 |
| | | 2 | 2 | 10.72 | 71.13 | 0.53 |
| | | 2 | 3 | 17.89 | 210.54 | 0.52 |
| | | 2 | 4 | 25.40 | 274.89 | 0.51 |
| | Stand | 2 | 2 | 12.10 | 145.37 | 0.53 |
| | | 2 | 3 | 18.70 | 72.27 | 0.49 |
| | | 2 | 4 | 24.20 | 318.57 | 0.54 |
| Coweeta | Down | 1 | 1 | 6.16 | 50.04 | 0.46 |
| | | 1 | 2 | 8.84 | 54.29 | 0.46 |
| | | 2 | 2 | 12.98 | 118.03 | 0.53 |
| | | 2 | 3 | 16.88 | 233.22 | 0.54 |
| | | 2 | 4 | 24.57 | 142.78 | 0.56 |
| | Stand | 2 | 2 | 12.47 | 131.50 | 0.53 |
| | | 2 | 3 | 18.10 | 188.99 | 0.52 |
| | | 2 | 4 | 24.41 | 176.84 | 0.52 |
| Duke | Down | 1 | 1 | 6.09 | 55.80 | 0.46 |
| | | 1 | 2 | 8.88 | 39.03 | 0.45 |
| | | 2 | 1 | 6.33 | 2.91 | 0.47 |
| | | 2 | 2 | 11.55 | 152.72 | 0.47 |
| | | 2 | 3 | 17.86 | 519.20 | 0.53 |
| | | 2 | 4 | 25.85 | 61.89 | 0.65 |
| | Stand | 2 | 2 | 12.79 | 95.07 | 0.55 |
| | | 2 | 3 | 18.34 | 337.59 | 0.52 |
| | | 2 | 4 | 27.73 | 115.41 | 0.52 |
| Fraser | Down | 1 | 1 | 5.97 | 56.84 | 0.46 |
| | | 1 | 2 | 8.71 | 25.12 | 0.45 |
| | | 2 | 2 | 12.05 | 123.11 | 0.54 |
| | | 2 | 3 | 17.71 | 190.58 | 0.54 |
| | | 2 | 4 | 26.35 | 224.18 | 0.57 |
| | Stand | 2 | 2 | 11.95 | 98.86 | 0.52 |
| | | 2 | 3 | 18.52 | 254.88 | 0.53 |
| | | 2 | 4 | 25.13 | 190.15 | 0.53 |
| Marcell | Down | 1 | 1 | 6.04 | 51.11 | 0.48 |
| | | 1 | 2 | 8.23 | 42.14 | 0.46 |
| | | 2 | 2 | 12.75 | 124.52 | 0.53 |
| | | 2 | 3 | 18.68 | 276.51 | 0.52 |
| | | 2 | 4 | 25.63 | 89.62 | 0.49 |
| | Stand | 2 | 2 | 12.72 | 168.25 | 0.53 |
| | | 2 | 3 | 17.64 | 162.53 | 0.54 |
| | | 2 | 4 | 24.88 | 104.46 | 0.61 |

(*Continued*)

**Table 2.** (Continued)

| Location | Position | Species | Size | Diameter (cm) | Log Mass (kg) | Wood Density (g cm$^{-3}$) |
|---|---|---|---|---|---|---|
| San Dimas | Down | 1 | 1 | 6.05 | 57.59 | 0.47 |
| | | 1 | 2 | 8.42 | 27.78 | 0.45 |
| | | 2 | 2 | 11.89 | 91.09 | 0.53 |
| | | 2 | 3 | 18.61 | 252.68 | 0.53 |
| | | 2 | 4 | 25.71 | 196.79 | 0.52 |
| | Stand | 2 | 2 | 11.80 | 59.63 | 0.54 |
| | | 2 | 3 | 18.14 | 351.20 | 0.55 |
| | | 2 | 4 | 24.77 | 96.27 | 0.56 |
| Santee | Down | 1 | 1 | 6.23 | 53.12 | 0.46 |
| | | 1 | 2 | 8.62 | 41.30 | 0.46 |
| | | 2 | 2 | 8.85 | 4.95 | 0.52 |
| | | 2 | 3 | 18.55 | 1101.15 | 0.53 |
| | | 2 | 4 | 24.42 | 319.22 | 0.54 |
| | Stand | 2 | 2 | 13.05 | 79.96 | 0.50 |
| | | 2 | 3 | 19.16 | 297.43 | 0.53 |
| | | 2 | 4 | 24.93 | 290.02 | 0.55 |
| Sierra Ancha | Down | 1 | 1 | 5.59 | 38.34 | 0.49 |
| | | 1 | 2 | 8.90 | 64.66 | 0.46 |
| | | 2 | 2 | 13.27 | 50.41 | 0.54 |
| | | 2 | 3 | 19.31 | 472.78 | 0.54 |
| | | 2 | 4 | 23.42 | 131.45 | 0.56 |
| | Stand | 2 | 2 | 13.08 | 108.58 | 0.54 |
| | | 2 | 3 | 20.16 | 282.15 | 0.55 |
| | | 2 | 4 | 25.26 | 310.22 | 0.56 |
| Tenderfoot Creek | Down | 1 | 1 | 6.22 | 47.35 | 0.46 |
| | | 1 | 2 | 8.76 | 54.62 | 0.47 |
| | | 2 | 2 | 12.31 | 102.97 | 0.56 |
| | | 2 | 3 | 18.98 | 297.57 | 0.54 |
| | | 2 | 4 | 24.80 | 190.69 | 0.55 |
| | Stand | 2 | 2 | 11.48 | 119.20 | 0.53 |
| | | 2 | 3 | 17.81 | 211.35 | 0.53 |
| | | 2 | 4 | 24.75 | 94.97 | 0.54 |

*: species: 1, hardwood; 2, softwood

size: 1, 4.5–7.5 cm; 2, 7.5–15.0 cm; 3, 15.0–22.5 cm; 4, 22.5–30.0 cm; 5, 30.0–37.5 cm; 6, 37.5–45.0 cm in diameter.

biomass, litter fall, and LAI varied with time; the increments were estimated by species group, and stand age.

The model was also parameterized for assessing CWD decomposition on WS80 at the Santee EF, where Hurricane Hugo left approximately 130 Mg ha$^{-1}$ in CWD (Table 3) based on the damage inventory conducted by Hook et al. [30]. Data for vegetation, climate, and soil used to parameterize the model for this application are similar to those used for assessing the impact of the hurricane on C sequestration in this watershed [33].

## Model performance evaluation

Model calibration and validation were evaluated using four widely employed quantitative methods [40]: the determination coefficient ($R^2$, squared Pearson's correlation coefficient),

**Table 3. CWD mass of each size class on WS80[*].**

| Size Class | Downed | | Standing | |
|:---:|:---:|:---:|:---:|:---:|
| | **Soft** | **Hard** | **Soft** | **Hard** |
| 1 | 0 | 0 | 0 | 0 |
| 2 | 3.90 | 7.30 | 0 | 0.50 |
| 3 | 11.21 | 11.98 | 0.50 | 1.01 |
| 4 | 20.90 | 17.190 | 2.49 | 1.02 |
| 5 | 23.80 | 12.00 | 3.49 | 1.01 |
| 6 | 10.20 | 0 | 1.50 | 0 |

[*]: Unit, Mg ha$^{-1}$; the total CWD mass on WS80 was 130 Mg ha$^{-1}$

size: 1, 4.5–7.5 cm; 2, 7.5–15.0 cm; 3, 15.0–22.5 cm; 4, 22.5–30.0 cm

5, 30.0–37.5 cm; 6, 37.5–45.0 cm in diameter.

model performance efficiency (E) [41], percent bias (PBIAS), and the RRS [the ratio of the root mean squared error (RMSE) to SD (standard deviation)] [42].

E ($-\infty$, 1) is the key variable used for evaluating the model performance and is calculated as

$$E = 1 - \frac{\sum (O_i - P_i)^2}{\sum (O_i - \bar{O})^2} \tag{1}$$

where $O_i$ and $P_i$ are the observed values and the predicted values obtained from the modelling, respectively; $\bar{O}$ is the observed mean. The evaluation criteria are: if E<0.0, the model is not applicable; $0.0 \leq E < 0.25$, the model performs poorly; $0.25 \leq E < 0.5$, the model performance is fair; $0.5 \leq E < 0.75$, model performance is good; and if $E \geq 0.75$, the model performs excellently.

The evaluation variables, PBIAS (criteria: between -25.0% and 25.0%) and RRS (criteria: 0.0–0.7), are calculated, respectively, as

$$PBIAS = \frac{\sum (O_i - P_i)}{\sum O_i} \times 100 \tag{2}$$

and

$$RRS = \frac{RMSE}{SD} \tag{3}$$

where SD is the observed standard deviation; RMSE is the root mean squared error between observed and simulated values, calculated as

$$RMSE = \sqrt{\frac{\sum (O_i - P_i)^2}{n}} \tag{4}$$

where $n$ is the number of samples or the pairs of the observed and simulated values.

The time to achieve 50% mass loss of CWD from the original mass (half-life, $T_{50}$) is a useful metric for comparing decomposition responses; it was calculated as

$$T_{50} = -Ln(0.5) \div k \tag{5}$$

where $k$ is the decomposition constant, y$^{-1}$, based on the single exponential model that is widely used to assess CWD decomposition as follows,

$$M_t = M_0 \times e^{-kt} \tag{6}$$

where $M_t$ is the mass remaining at time $t$ (years); $M_0$ is the initial mass; $k$ is the CWD decomposition constant.

Because the CWD decomposition may not follow a perfect exponential model [3, 13, 21, 43], a function combining power and exponential components (Eq 7) was used to assess the CWD decomposition, i.e.,

$$M_t = M_0 \times (t+1)^{k_2} \times e^{-k_1 t} \tag{7}$$

where $k_1$ and $k_2$ are the CWD decomposition constants. Since Eq 7 has two decomposition constants, Eq 5 cannot be directly used to compute the half-life of CWD; accordingly, $T_{50}$ for this function was calculated using an iterative method [44].

## Results

### Model calibration

Four metrics for evaluating model performance for the calibration step were used to determine if the model is in the line of the design objectives. The PBIAS resulted from model calibration was 10.3 for MLML, and 7.39 for the CLML (Table 4). RRS was 0.51 and 0.35 for calibration against the MLML and CLML respectively. These metrics were within acceptable ranges [PBIAS $\in$ (-25.0, 25.0) and RRS $\in$ (0.0, 0.7)]. The determination coefficient $R^2$ (Fig 3) showed that the simulated mass loss was highly correlated with both the MLML ($R^2 = 0.80$) and CLML ($R^2 = 0.90$). The simulated total mass loss fraction (0.331) over six years ($T_0$ to $T_6$) for model calibration was about 10.3% lower than the MLML (0.369), and the simulated total mass loss fraction (0.416) over six years was 7.4% lower than the CLML (0.449). The difference in mass loss fraction between the MLML and CLML reflect uncertainties in estimating log mass loss.

Model performance efficiency [E = 0.71, E $\in$ (-$\infty$, 1.0)] for the calibration against the measured mass loss showed that the model performance was within the rating range of "Good" (0.50$\leq$E<0.75). The efficiency (E = 0.87) of model calibration for the calculated mass loss using the disks collected *in situ* showed that the model performance was excellent (E$\geq$0.75), and reflects a difference in model performance efficiency for model calibration against data obtained from different methods (see Discussion below). These performance metrics from model calibration step indicate that the model is applicable for assessing CWD decomposition across the range conditions on the EFs, and it is in line with design objectives [29].

### Model validation

Similar to the model performance evaluation for the model calibration, the four model evaluation metrics provide a basis for assessing the validation step, which affirms the applicability of

**Table 4. Results from the model performance evaluation\*.**

| Item | Calibration | | Validation | |
|---|---|---|---|---|
| | Measured | Calculated | Measured | Calculated |
| E | 0.707 | 0.868 | 0.818 | 0.734 |
| $R^2$ | 0.801 | 0.900 | 0.822 | 0.758 |
| PBIAS | 10.30 | 7.391 | 1.013 | -4.627 |
| RRS | 0.510 | 0.348 | 0.404 | 0.498 |
| Mean | 0.369 | 0.449 | 0.364 | 0.359 |
| Predicted mean | 0.331 | 0.416 | 0.360 | 0.375 |

\*: Measured, mass loss measured in situ; Calculated, mass loss calculated based on the disks collected from experimental logs *in situ*; Mean, mass loss fraction measured or calculated; predicted mean, mass loss fraction from simulation.

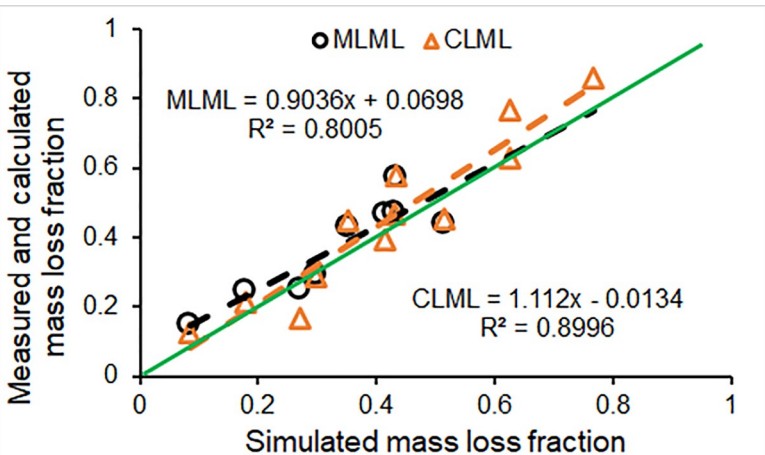

**Fig 3. Results from model calibration: Comparing the simulated log mass loss fraction to the measured loss fraction *in situ* (MLML) and the calculated loss fraction based on the disk wood density loss (CLML); black dashed line, regressed relationship between the measured mass loss and the simulated loss; orange dashed line, regressed correlation between the calculated mass loss and the simulated loss; green line, 1:1.**

the model to the study. Model validation against measured and calculated mass loss of logs from the FWDE were 1.01 and -4.63 for PBIAS, 0.40 and 0.50 for RRS, and 0.82 and 0.76 for $R^2$ (Fig 4; Table 4), respectively. These metrics were within acceptable ranges for the metrics [PBIAS $\in$ (-25.0, 25.0), RRS $\in$ (0.0, 0.7) and $R^2 \in$ (0.0, 1.0)]. The simulated fraction of total mass loss (0.360) over the six-year simulation period was 1.0% lower than the MLML (0.364), but the simulated mass loss fraction (0.375) was approximately 4.6% higher than the CLML (0.359). The model performance efficiency (E) was 0.82 and 0.74 for the validation against the measured and calculated mass loss values, respectively; indicating that model performance was within the rating range of "Excellent" (E≥0.75) when validated against measured mass loss and within the range of "Good" when validated against calculated mass loss (E = 0.74, between 0.5 and 0.75).

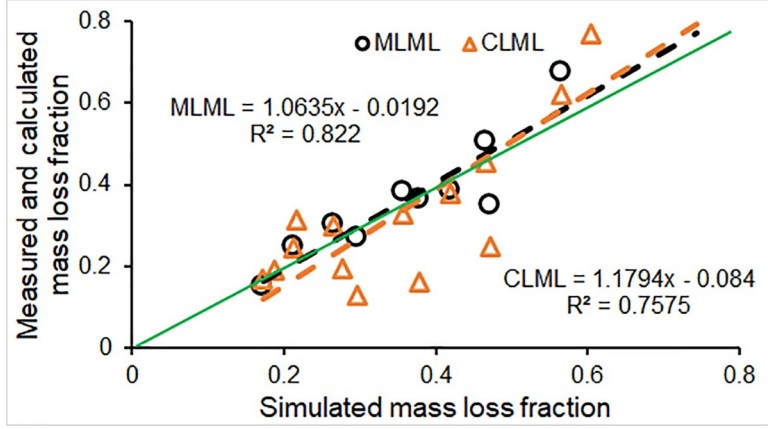

**Fig 4. Results from model validation: Comparing simulated log mass loss fraction to measured loss fraction *in situ* (MLML), and the calculated loss fraction based on the disk wood density loss (CLML); black dashed line, regressed relationship between the measured mass loss and the simulated loss; orange dashed line, regressed relationship between the calculated mass loss and the simulated loss; green line, 1:1.**

## Application

To further test the model and demonstrate a practical application, we simulated CWD decomposition following the massive biomass blow-down following Hurricane Hugo in 1989. Our simulations tracked the rapid decomposition of the CWD with an estimated 1.6% of the original CWD mass (130 Mg ha$^{-1}$) following the Hurricane Hugo remaining 30 years after the storm. The decomposition pattern followed an exponential function (Fig 5), with a decomposition constant of 0.132 y$^{-1}$ based on the single exponential model with an intercept forced to 100% of the initial CWD mass (100%), and about 0.139 y$^{-1}$ based on the exponential model of the simulated decomposition without forcing the intercept. However, the R$^2$ (squared Person's correlation coefficient) was high (>0.99) for both exponential equations with and without forcing the intercept (Fig 5).

The CWDDAT also accounts for transfer of CWD-C into the forest floor and mineral soil. A small amount of fine woody debris (0.056% of original mass) derived from CWD decomposition remained on the forest floor, and a small amount of C was incorporated into the mineral soil due to leaching. The predicted DOC incorporated into soils over the 30 years was 30.3 g C m$^{-2}$ in total, accounting for 0.474% of the total CWD resulting from the hurricane. The predicted particulate organic carbon (POC) incorporated into the mineral topsoil was 0.13 g C m$^{-2}$ in total, only about 0.002% of total C loss to the CWD decomposition over 30 years.

Decomposition of CWD in WS80 was primarily mediated by fungi, accounting for approximately 72.0% of the total CWD mass loss over the last 30 years. Termites, another key decomposer, accounted for approximately 24.5% of the total CWD mass loss. The contributions of other decomposers (including bacteria and beetles) to CWD decomposition were small, accounting for approximately 3.5% of the total mass loss.

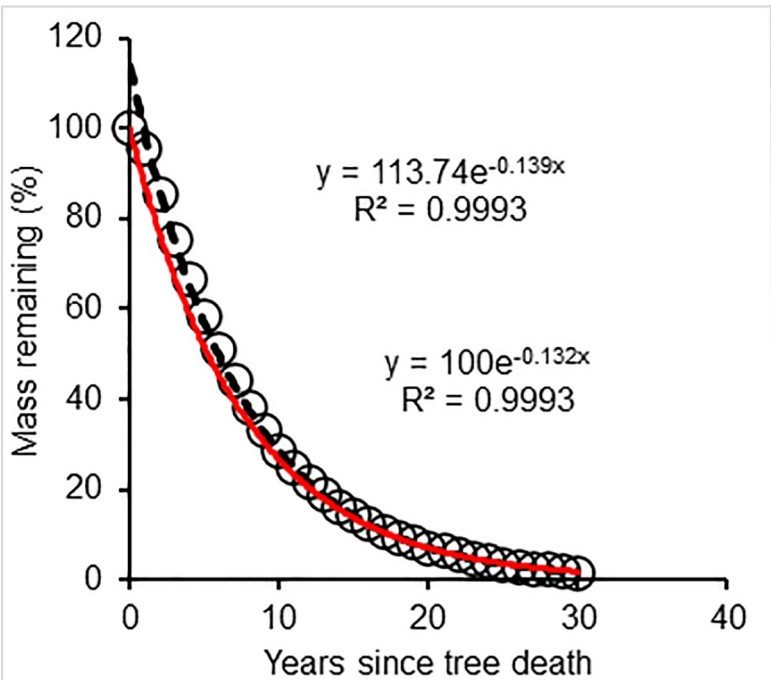

**Fig 5. Mass attenuation of CWD on WS80 watershed after Hurricane Hugo on the Santee Experimental Forest, South Carolina due to decomposition; black dashed line is regressed without forcing the intercept to 100% of the initial wood mass; red dashed line is regressed with forcing the intercept to100% of the initial wood mass; the initial wood mass was 130 Mg ha$^{-1}$, based on the damage inventory [30, 35].**

## Discussion

### Model calibration and validation

The four model evaluation metrics consistently showed that the CWDDAT model is applicable for assessing CWD decomposition in forests across a wide bioclimatic range. However, those metrics varied depending on the basis for comparison; for example, the calibration performance efficiency with the measured CWD mass loss (E = 0.71) was less than the calculated CWD mass loss (E = 0.87). This difference in model performance might be related to the errors between measurement and calculation (Fig 6). The predicted means for model calibration were 10.8 and 8.1% lower than the MLML and CLML, respectively. The simulated mean for validation was 2.1% lower than the measurement, but 3.6% higher than the calculated value. The predicted average wood mass loss from all samples, including those used to calibrate and validate the model, was 5.4% lower than the measured value, and 1.4% lower than the calculated mean, reflecting that the model slightly underestimates CWD decomposition rates.

Since the simulation results are compared against two different estimates of mass loss, the differences in model performance metrics as compared to the "measured" or "calculated" mass loss essentially represent uncertainty in the estimate of mass loss. The important aspect to convey is that the model performance was good, regardless of the source of the mass loss estimate. Fig 6 is important in showing the uncertainties between the "measured" and "calculated" mass loss estimates; it shows that generally mass loss "measured" > "calculated". However, importantly, the model did a fine job predicting decomposition across a wide range of mass loss over the 6 years (differences in mass loss fraction among the logs: <0.02–0.78).

The differences in model performance relative to the basis for comparison reflects both uncertainties within the model algorithms and in the variables used for comparison. The challenge in assessing wood decomposition is to obtain a metric (e.g., mass, wood density) that is representative of the log at each time-step assessed. Since the FWDE is a long-term study, it was not possible to destructively sample entire logs, so weighing the entire log was used to overcome some of the limitations associated with only using samples from the end-of-log disks to represent the log decomposition. However, the assumption that the moisture content from the disk samples represented the log is also problematic, since the disk has one surface

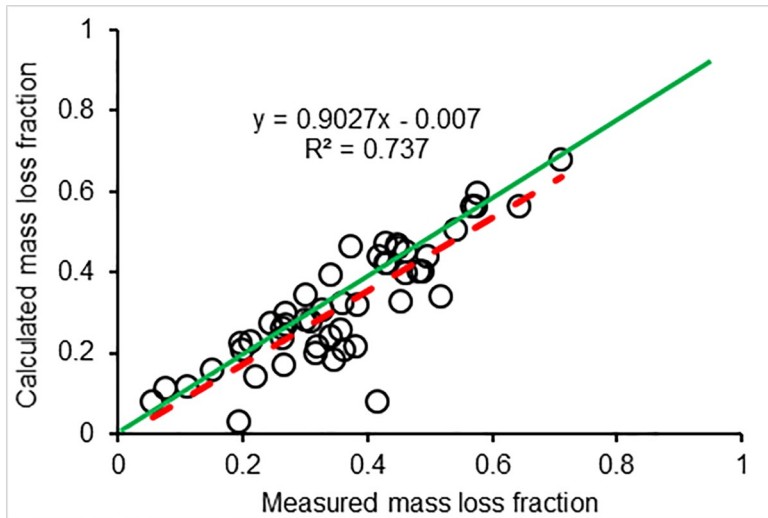

**Fig 6. Measured vs calculated wood mass loss based on the disks collected from FACE logs in the nine EFs; red dashed line, regressed relationship between the measured mass loss and calculated loss; green line, 1:1.**

exposed. Accordingly, if the sample moisture content were less than the mean for the entire log, the measured log mass would be overestimated. Analogous issues arise when results from field incubation studies are used to infer decomposition of CWD [2, 11, 22]. However, it's highlighted in this assessment when to two, albeit not independent, metrics are used as the basis for model comparison.

Reassuringly, both the model calibration and validation showed that the model performed well for assessing CWD decomposition across a wide range in climatic conditions, forest types and soil in the continental USA. However further assessment of the CWDDAT is warranted to affirm that it is robust across all forest lands. The current assessment utilizes relatively small logs, especially in comparison to the large trees typical of the tropics or pacific north west region. Accordingly, trials that focus on large diameter material is suggested. Similarly, it well established that wood decomposition rates vary among species [19, 22, 45–47], hence testing the model against other species is needed, especially for the dense tropical species. Similarly, testing across the highly variable boreal zone is warranted [5, 48, 49]. Overall, this work demonstrates that the CWDDAT [29] is a useful tool for assessing CWD decomposition.

## Model application

CWD decomposes quickly in warm and humid locations. Accordingly, the fast CWD decomposition at WS80 reflects the subtropical forest in South Carolina, and the presence of subterranean termites. As a result, over 50% of CWD produced by Hurricane Hugo in 1989 had decomposed within 6 years ($T_{50} \leq 6$). The predicted main contributor of CWD decomposition on Santee is fungi, accounting for about 72% of total CWD mass loss, as the warm and humid climate regime is good for fungal growth [2]. Termites contributed about 24.5% of the total wood mass loss because Santee is located in the area with the highest risk of termites in USA [36].

Although the estimated mean CWD mass loss from the wood decomposition on WS80 followed an exponential decomposition function, there was a small difference in the decomposition constant ($k$) with and without forcing the intercept of the regressed function to the initial wood mass; forcing the intercept to 100% of the initial mass at T0 seems appropriate to have the function reflect the actual starting condition. The effect of altering the intercept changes the half-life ($T_{50}$) from 5.0 to 5.2 years based on the decomposition constants from regressed exponential equations with and without forcing the intercept, respectively, reflecting that there is a small time-lag in this subtropical climate regime. However, when Eq 7 was used to assess CWD decomposition, the dual constants $k_1$ and $k_2$ were 0.1535 ($k_1$) and 0.1303 ($k_2$), respectively, and $T_{50}$ was approximately 6.2 years. Accordingly, the simulated CWD decomposition constant on WS80 was within the global range of CWD decomposition and within the range in North America [25, 50]. The CWD decomposition constant from this study was also similar to that reported for other studies from similar climatic conditions [50–52].

The temporal contributions of decomposers to the mass loss of the CWD in WS80 were substantially different; the estimated contribution of termites was approximately 72% within the second year, and their contribution decreased rapidly to 28.0% of the total annual decomposition 2 years after the hurricane and their proportion of total decomposition continued to decline after year 2. Similar to termites, the contribution of beetles to the CWD decomposition decreased over time ranging from 5.3% to null. In contrast, the contributions of fungi to decomposition increased with time. Fungal respiration increased from 21.6% of the total annual mass loss within the first year to ≥70% after one year, and remained over 70% of the total annual wood mass loss until the end of the simulation time period (30 years). Bacterial respiration increased from 0.4% in the first year to 3.5% of the total annual mass loss in 2017, reflected that bacterial contribution to CWD decomposition was small overall [53].

The decayed logs remained the primary pool of CWD (average 98.1 g C m$^{-2}$) after 30 years. The predicted amount of fragmentation during CWD decomposition was approximately 23.5 g m$^{-2}$ y$^{-1}$, with approximately 7.3 g m$^{-2}$ remaining in the forest floor at the end of 30 years. The large difference in the mass of fragments produced and that remaining on the floor is due to decomposition of the fragmented material within the forest floor. There was only a small amount of wood C as POC (0.13 g C m$^{-2}$) from CWD decomposition, which was incorporated into the soil as a result of leaching.

DOC from CWD decomposition was the primary pathway for C incorporation into the soil [54]. The 30-year predicted total DOC from the CWD produced by Hurricane Hugo was about 35.5 g C m$^{-2}$, and the net incorporation was about 30.3 g C m$^{-2}$, a rate of approximately 1.01 g C m$^{-2}$ y$^{-1}$ on average, reflecting that DOC is the main contributor of wood C into the underlaying mineral soil. DOC incorporation into soil changed non-linearly over time, increasing from 0.16 g C m$^{-2}$ y$^{-1}$ in the first year to 5.9 g C m$^{-2}$ y$^{-1}$ in the 10$^{th}$ year after the hurricane. Transfer by DOC decreased substantially after year 10, from 5.9 g C m$^{-2}$ y$^{-1}$ in the 10$^{th}$ year to 0.09 g C m$^{-2}$ y$^{-1}$ at the 26$^{th}$ year due to a reduction in CWD decomposition. The DOC incorporated into soils can be absorbed or aggregated and therefore maintained in soils for a longer time [55, 56]. These simulation results provide context for suggesting the magnitude of the transfers of C from CWD into the soil. Those predictions are predicated on functional relationships [29], but those functions warrant testing through field studies.

This application demonstrates the utility of the model for assessing CWD dynamics, in this case following a large episodic event. Similarly, the model could be used to assess the annual turnover of CWD in forest stands affected by different management regimes or climate scenarios. The fine performance of the model across contrasting forest landscapes suggests the opportunity to conduct large-scale assessments of CWD dynamics. The availability of national forest inventory data that includes periodic assessments of CWD stocks (e.g., Forest Inventory and Analysis data in the U.S.) suggests an opportunity to assess the model performance among multiple forest types within or among regions. Such a large-scale assessment would further test the model, and if successful, provide a tool that could be used in conjunction with forest biomass simulators to fully assess the dynamics of forest biomass.

## Conclusions

The CWDDAT model used to assess CWD decomposition in forests was evaluated against wood mass loss observed from nine sites with different ecological conditions across the continental USA. Four quantitative model evaluation methods consistently demonstrated that the model is an effective tool for assessing CWD decomposition and the associated C fluxes in forests. Simulating CWD decomposition following Hurricane Hugo also demonstrated that this model is applicable for simulating responses at the watershed-scale. CWDDAT was built to reflect biogeochemical processes mediating wood decomposition; the calibration, validation and application simulation results demonstrate that those processes are sensitive to the primary factors known to regulate wood decomposition. The CWDDAT model incorporates both micro-organisms and arthropods as mediators of CWD decomposition, providing a basis for assessing the contributions of decomposer communities.

The CWDDAT model provides the basis for simulating the C cycle associated with forest CWD. The tool has considerable utility for assessing functional linkages between CWD decomposition and soil C, thereby providing a basis to the interactions of forest management, disturbance regimes (e.g., tropical storms), or climate change on this important component of the forest C cycle. Further evaluations of the model are warranted, and we hope that the

findings reported here stimulate further studies to further strengthen the basis for model development and applications.

## Supporting information

**S1 Table. Observed climate data at FWDE sites.**
(XLSX)

## Acknowledgments

The FWDE would not have been possible without the collaboration of Ram Oren, Duke Univ. who provided the loblolly pine logs and Andrew Burton, Michigan Technological Univ., who provided the aspen and birch logs. Installation and maintenance of the FWDE sites is attributable to Julie Arnold (Santee EF) with essential help from Joanne Tirocke (Moscow, ID), Helen Smith (Tenderfoot EF), Nate Aspelin (Marcel EF), Banning Starr (Fraser EF), Mike Oxford (San Dimas EF), Elizabeth Keppeler (Caspar Creek EF), and Dan Neary and Jackson Leonard (Sierra Ancha EF). We also thank Ben Bright, USDA Forest Service, Rocky Mountain Research Station, Moscow ID for producing the map. We acknowledge the useful comments of the reviewers of this manuscript.

## Author Contributions

**Conceptualization:** Zhaohua Dai, Carl C. Trettin, Andrew J. Burton, Martin F. Jurgensen, Brian T. Forschler, Daniel L. Lindner.

**Data curation:** Zhaohua Dai, Carl C. Trettin, Andrew J. Burton, Deborah S. Page-Dumroese, Brian T. Forschler, Jonathan S. Schilling.

**Formal analysis:** Carl C. Trettin, Deborah S. Page-Dumroese, Daniel L. Lindner.

**Funding acquisition:** Zhaohua Dai, Carl C. Trettin, Andrew J. Burton, Martin F. Jurgensen, Jonathan S. Schilling.

**Investigation:** Carl C. Trettin, Andrew J. Burton, Deborah S. Page-Dumroese, Brian T. Forschler, Jonathan S. Schilling, Daniel L. Lindner.

**Methodology:** Zhaohua Dai, Carl C. Trettin, Andrew J. Burton, Martin F. Jurgensen, Deborah S. Page-Dumroese, Brian T. Forschler, Jonathan S. Schilling.

**Project administration:** Carl C. Trettin, Andrew J. Burton.

**Software:** Zhaohua Dai.

**Supervision:** Carl C. Trettin, Andrew J. Burton.

**Validation:** Zhaohua Dai, Carl C. Trettin.

**Writing – original draft:** Zhaohua Dai.

**Writing – review & editing:** Carl C. Trettin, Andrew J. Burton, Martin F. Jurgensen, Deborah S. Page-Dumroese, Brian T. Forschler, Jonathan S. Schilling, Daniel L. Lindner.

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
