## [Decision Letter · Decision Letter 0]

16 Mar 2021

PONE-D-21-02848

Coarse Woody Debris Decomposition Assessment Tool: Model Validation and Application

PLOS ONE

Dear Dr. Dai,

Thank you for submitting your manuscript to PLOS ONE. After careful consideration, we feel that it has merit but does not fully meet PLOS ONE’s publication criteria as it currently stands. Therefore, we invite you to submit a revised version of the manuscript that addresses the points raised during the review process.

We look forward to receiving your revised manuscript.

Kind regards,

Dafeng Hui, Ph.D.

Academic Editor

PLOS ONE

Additional Editor Comments:

I now have one report from an expert review who finds the manuscript has many merits, but also raises some technique concerns, such as a relatively short term study for wood decomposition and a lack of detailed decomposition rate data. The manuscript needs to be sustainably revised before it can be accepted for publication.

Journal Requirements:

2. In your Methods section, please provide additional information regarding the permits you obtained to collect samples for the present study. Please ensure you have included the full name of the authority that approved the field site access and, if no permits were required, a brief statement explaining why.

"Funding to install the FWDE was provided by the USDA Forest Service and Michigan

Technological Univ. The model development was supported by the US Dept. of Energy (DE352

SC0016235), US NSF (DEB 1754603) and the USDA Forest Service."

"Please see Acknowledgments "

"NO"

7. We note that Figures 1 and 2 in your submission contain map images which may be copyrighted. All PLOS content is published under the Creative Commons Attribution License (CC BY 4.0), which means that the manuscript, images, and Supporting Information files will be freely available online, and any third party is permitted to access, download, copy, distribute, and use these materials in any way, even commercially, with proper attribution. For these reasons, we cannot publish previously copyrighted maps or satellite images created using proprietary data, such as Google software (Google Maps, Street View, and Earth). For more information, see our copyright guidelines: http://journals.plos.org/plosone/s/licenses-and-copyright.

7.1.    You may seek permission from the original copyright holder of Figures 1 and 2 to publish the content specifically under the CC BY 4.0 license. 

7.2.    If you are unable to obtain permission from the original copyright holder to publish these figures under the CC BY 4.0 license or if the copyright holder’s requirements are incompatible with the CC BY 4.0 license, please either i) remove the figure or ii) supply a replacement figure that complies with the CC BY 4.0 license. Please check copyright information on all replacement figures and update the figure caption with source information. If applicable, please specify in the figure caption text when a figure is similar but not identical to the original image and is therefore for illustrative purposes only.

Reviewers' comments:

Reviewer's Responses to Questions

**Comments to the Author**

1. Is the manuscript technically sound, and do the data support the conclusions?

Reviewer #1: Partly

2. Has the statistical analysis been performed appropriately and rigorously? 

Reviewer #1: Yes

3. Have the authors made all data underlying the findings in their manuscript fully available?

Reviewer #1: Yes

4. Is the manuscript presented in an intelligible fashion and written in standard English?

Reviewer #1: Yes

5. Review Comments to the Author

Reviewer #1: General comments

I found many positive aspects to this manuscript. We absolutely need to be able to predict how dead wood pools respond to climate change and management. We need to move to mechanistic models. We need broad-scale experiments to calibrate and test these models. The current manuscript describes a study in which this has been done and while it is not the end of this process it certainly provides an interesting and very good example of what could be done.

However, the authors seem to gloss over the shortcomings of their work. Don’t get me wrong, for the resources available and the state of the science in this field they did a very nice job. But decomposition of woody debris can take decades to centuries, so a 6 year study is short even for the tropics. Having data at one point in time is better than no data, but it does place some real limitations on the kinds of calibration and predictions possible. Although their experiment and sites spanned a wide range of conditions, they are somewhat limited to those in which woody debris is decomposing within. Again, not the fault of the authors to study as much as they could give resources, etc. But it needs to be pointed out that colder and warmer sites exist globally as well as drier and wetter ones. They largely worked under forest canopies, but after severe disturbances the dead wood might be decomposing in open settings that their experiment did not consider. They tested three species belonging to widespread genera; however, none of these species had strong decay resistance. The experiment likely under estimated the range of decomposition possible within a site. So while I understand that one can only do so much, it would be useful to acknowledge this and then suggest how these limitations might be overcome. Or which ones should be given priority. This would form the basis of a very useful discussion.

There are various results in the manuscript that don’t seem to have any methods. I realize that the emphasis is on the model. However, there are many detailed results such as the increase in soil stores of C from CWD decomposition that don’t seem to have any specific methods. Same as for the proportion of decomposition attributed to different causes. All very interesting, but unclear as to how the numbers were derived.

Oddly there is not very much regarding the decomposition rates themselves. It might be useful to start the results section not with model calibration, but with a quick review of the decomposition results. I understand the main focus is on the model, but the calibration and validation do depend on the underlying decomposition data.

I found the discussion to largely be either a rehash of the results or to present what seems to be new results. There is very little actual discussion. In the conclusions there is a sentence or two on the need for more data. That is good and obviously true. But in the discussion the kinds of data or studies could be developed so that this statement has some basis. For example, are longer studies needed? Or more species or places or conditions? Is there a need to improve process level studies to provide better insights into the actual mechanisms? Addressing these issues might lead to a more interesting and useful discussion.

The manuscript constantly switches from decay to decomposition to decay. This is fairly confusing and unnecessary. I urge the authors to choose one term and stick with it. I prefer to call the process decomposition and the state it reaches decay. However, the authors can use either, but they should be consistent.

Below are a number of specific comments that should be addressed to improve the clarity of the manuscript.

Specific comments (line)

14 This sentence has two problems. The first is the grammar is incorrect. It should use the word termed instead of term. But it is also factually incorrect. Although there is no set diameter cutoff, there is fine woody debris as well as coarse woody debris. Also woody debris contains woody materials such as bark was well as wood. This is not a good way to start an abstract.

24 Is there some numerical indication of this degree of accuracy?

77 Although this is a review of another study, I have to remark that some of the factors mentioned are clearly confounded. Climate is related to location as is altitude. Hopefully these were not all used simultaneously.

79 I am not sure viable is the best word to use here. I suggest switching it to possible. I think the purpose of the current study is to test viability. The other study seems to suggest it is a possible model to be evaluated.

81-83 I find the wording is odd in this section. I don’t see how model calibration affirms the design. Usually the first step in testing a model is to verify the various functions and responses to show they are as they should be. Perhaps this was taken care of in the earlier study. I guess one could then move on to whether the model can be calibrated to field data? I am not sure what mode availability is. Perhaps model availability? But even then I don’t know how “validating” indicates availability. I gather this was a corroboration phase in which one tests the model predictions to an independent data set. But it is worded in a most odd manner.

100 Is it possible to destroy an ecosystem short of a nuclear blast? Perhaps this could be reworded to be less value laden and more accurate? For example, I could see that a large share of the trees were felled by the storm. I can’t see that they were destroyed because then there would be no need to study the decomposition because the trees even in dead form would not be there.

121 I have concerns that only one disk was removed. The moisture content and extent of decay would likely vary with distance to the ends. Where was the disk removed from? A standard distance from the end? What was that? This is a critical detail.

128 This does not entirely seem correct. Validation as described seems to be what I would call verification. Validation or corroboration is a test of a model’s ability to predict against an independent data set. Maybe the authors are just following the terms laid out by Oreskes et al, but if that is the case I find the terms a bit of a muddle.

192 What I am not understanding is how one could possibly apply a non-linear model or a complex model to one data point in time. I think one is basically stuck with the linear negative exponential model if one has one point in time (or two if you count the initial value).

198 It would be helpful if this paragraph started with a topic sentence summarizing the results presented in the paragraph.

205 It is not entirely clear what the numbers in the parentheses are referring to. Is this the fraction remaining? Or lost?

212 I am not sure what to make of the statement “satisfies users’ basic expectations”. Wouldn’t that depend on the user and their expectations? It seems to imply some ability to read minds. Perhaps this could be reworded?

218 Again not sure what the numbers in the parentheses are referring to. Is this the total fraction mass lost?

238 While this is a very interesting result and don’t question it, I do have to wonder about how the number was derived. Was this part of another study? Or if it is part of this study’s analysis: a few more details should be provided. For example, was the C concentration under logs monitored to determine the increase? How was the soil sampled? How was C concentration determined? To what depth? How was the change under logs scaled to the broadscale? Or was this a result of the models predictions?

253 I am not following the logic here. It might be a matter of wording.

259 The details of disk location should have been in the methods. If one really wanted a good sample of conditions one should have removed disks at several locations so that a log level average could be derived.

292-297 This seems more along the lines of results. Perhaps it needs to be set up better as a discussion point? Perhaps if it started “ A benefit of a mechanistic model is that it can provide insights into the basic processes causing decomposition”? Then go into the mechanisms that are suggested?

504 A Y-intercept above 100% indicates the presence of a time-lag. Forcing the negative exponential through 100% artificially eliminates this time lag. A better method is to use a model that includes a time lag to get the intercept to be 100%. There are various models that can do this.

508 Was there any monitoring of the CWD mass in the watersheds? Or was there just an estimate of the initial input? Or was this just a prediction of the model that can’t be checked because the CWD was not repeatedly inventoried?

517 Did the logs contain any bark? Or were they all wood? Or was bark considered to be part of the wood? Was this the initial wood density or the average wood density over the course of the experiment? I suspect the former, but one can’t tell from what is written. I may have missed it, but I did not see any size 5 or 6 logs in the table. Why define size classes that were not used?

6. PLOS authors have the option to publish the peer review history of their article (what does this mean?). If published, this will include your full peer review and any attached files.

Reviewer #1: No

---

## [Author Response · Author response to Decision Letter 0]

29 May 2021

Dear Editor:

Response: Thank you. The file of "responses to editor and reviewers" is uploaded.

Response: Thank you. We uploaded the revised version with track changes as you suggested.

Response: Thank you. We did as you suggested.

Sincerely,

Zhaohua Dai

---

## [Editor Report · Decision Letter 1]

28 Jun 2021

Coarse Woody Debris Decomposition Assessment Tool: Model Validation and Application

PONE-D-21-02848R1

Dear Dr. Dai,

We’re pleased to inform you that your manuscript has been judged scientifically suitable for publication and will be formally accepted for publication once it meets all outstanding technical requirements.

Kind regards,

Dafeng Hui, Ph.D.

Academic Editor

PLOS ONE

Additional Editor Comments (optional):

The authors have made efforts and adequately addressed the reviewer's concerns.
---

## [Editor Report · Acceptance letter]

1 Jul 2021

PONE-D-21-02848R1 

Coarse Woody Debris Decomposition Assessment Tool: Model Validation and Application 

Dear Dr. Dai:

I'm pleased to inform you that your manuscript has been deemed suitable for publication in PLOS ONE. Congratulations! Your manuscript is now with our production department. 

Kind regards, 

on behalf of

Dr. Dafeng Hui 

Academic Editor

PLOS ONE